Brain tissue oxygen pressure combined with intracranial pressure monitoring may improve clinical outcomes for patients with severe traumatic brain injury: a systemic review and meta-analysis

Zhang Chengcheng 1 2
Zhou Lingmin 1 3
Zhang Kai 1
Huang Jian 4
Cao Lanxin 1
Lou Yuhang 1
Fan Yushi 1
Zhang Xinyun 1
Wang Yesong 1
Cui Wei 1
Hu Lihua hhx1819@sina.com 5
Zhang Gensheng genshengzhang@zju.edu.cn 1 6
1 Department of Critical Care Medicine, Second Affiliated Hospital, Zhejiang University School of Medicine , Hangzhou , China
2 Department of Emergency Medicine, Affiliated Hospital of Hangzhou Normal University , Hangzhou , China
3 Department of Critical Care Medicine, First People’s Hospital of Taizhou , Taizhou , China
4 Department of Critical Care Medicine, Hangzhou Ninth People’s Hospital , Hangzhou , China
5 Department of Critical Care Medicine, Hospital of Zhejiang People’s Armed Police , Hangzhou , China
6 Key Laboratory of Multiple Organ Failure (Zhejiang University), Ministry of Education , Hangzhou , China
Abdullah Jafri
Electronic publication date: 2024 Oct 8
Publication date: 2024
Volume: 12
Electronic Location ID: e18086
Received 2024 Jun 7; Accepted 2024 Aug 21
Copyright: ©2024 Zhang et al.
Copyright year: 2024
Copyright holder: Zhang et al.
License: This is an open access article distributed under the terms of the Creative Commons Attribution License, which permits unrestricted use, distribution, reproduction and adaptation in any medium and for any purpose provided that it is properly attributed. For attribution, the original author(s), title, publication source (PeerJ) and either DOI or URL of the article must be cited.
License URL: https://creativecommons.org/licenses/by/4.0/

Keywords: Brain tissue oxygen pressure, Intracranial pressure, Severe traumatic brain injury, Meta-analysis

Funding: National Natural Science Foundation of China No.82100012 This work was supported by grants from the National Natural Science Foundation of China (No.82100012, Kai Zhang). The funders had no role in study design, data collection and analysis, decision to publish, or preparation of the manuscript.

==============================
Background

Although the optimization of brain oxygenation is thought to improve the prognosis, the effect of brain tissue oxygen pressure (PbtO2) for patients with severe traumatic brain injury (STBI) remains controversial. Therefore, the present study aimed to determine whether adding PbtO2 to intracranial pressure (ICP) monitoring improves clinical outcomes for patients with STBI.

Methods

PubMed, Embase, Scopus and Cochrane Library were searched for eligible trials from their respective inception through April 10th, 2024. We included clinical trials contrasting the combined monitoring of PbtO2 and ICP versus isolated ICP monitoring among patients with STBI. The primary outcome was favorable neurological outcome at 6 months, and secondary outcomes including the in-hospital mortality, long-term mortality, length of stay in intensive care unit (ICU) and hospital.

Results

A total of 16 studies (four randomized studies and 12 cohort studies) were included in the meta-analysis. Compared with isolated ICP monitoring, the combined monitoring was associated with a higher favorable neurological outcome rate at 6 months (RR 1.33, 95% CI [1.17–1.51], P < 0.0001, I2 = 0%), reduced long-term mortality (RR 0.72, 95% CI [0.59–0.87], P = 0.0008, I2 = 2%). No significant difference was identified in the in-hospital mortality (RR 0.81, 95% CI 0.66 to 1.01, P = 0.06, I2 = 32%), length of stay in ICU (MD 2.10, 95% CI [−0.37–4.56], P = 0.10, I2 = 78%) and hospital (MD 1.07, 95% CI [−2.54–4.67], P = 0.56, I2 = 49%) between two groups. However, the pooled results of randomized studies did not show beneficial effect of combined monitoring in favorable neurological outcome and long-term mortality.

Conclusions

Currently, there is limited evidence to prove that the combined PbtO2 and ICP monitoring may contribute to improved neurological outcome and long-term mortality for patients with STBI. However, the benefit of combined monitoring should be further validated in more randomized studies.

Background

Traumatic brain injury (TBI) represents a significant and potentially fatal condition on a global scale, boasting the highest occurrence among prevalent neurological disorders (Maas et al., 2022). As reported by the Lancet Neurology Commission in 2017, TBI’s enduring status as a leading contributor to injury-related mortality and disability is projected to persist until 2030 (Maas et al., 2017). Furthermore, severe traumatic brain injury (STBI), characterized by an initial Glasgow Coma Scale (GCS) score below 9 upon admission, presents a dire prognosis, resulting in mortality for 25 to 40% of afflicted individuals (Maas et al., 2022). Mechanical trauma leads to primary brain injury, resulting in tissue damage and distortion shortly after the injury. Secondary injuries following TBI alter cellular function and propagate the trauma through mechanisms such as free-radical production, depolarization, edema formation, excitotoxicity, blood–brain barrier disruption, calcium imbalance, and intracranial hematoma (De Macedo Filho et al., 2024). Despite notable clinical and structural variability in STBI manifestations, consensus persists regarding the paramount importance of mitigating secondary brain injury in TBI management. A proper monitoring of the injured brain is necessary to adapt to the treatment according to the specific status of the patients (Kaur & Sharma, 2018).

Early identification of secondary brain damage relies on neuromonitoring in patients with TBI. Given that intracranial hypertension stands as an autonomous risk factor for both mortality and neurological impairments (Stocchetti et al., 2017), global guidelines underscore the necessity of intracranial pressure (ICP) surveillance (Hawryluk et al., 2022) following severe traumatic brain injury (Carney et al., 2017; Geeraerts et al., 2018). Compared with no ICP monitoring, the implementation of ICP monitoring correlates with heightened therapy engagement, diminished mortality rates, and improved functional outcomes six months post-injury (Robba et al., 2021). Nevertheless, the potential for secondary insults persists even in the absence of elevated ICP levels (Dobrzeniecki et al., 2021), prompting the recommendation for advanced cerebral monitoring within neurocritical care settings (Kaur & Sharma, 2018). Brain tissue hypoxia can manifest independently of elevated intracranial pressure and may independently correlate with adverse neurological outcomes. Consequently, brain tissue oxygen pressure (PbtO2) has surfaced as a promising avenue, enabling the identification of critical thresholds associated with neuronal tissue infarction (Gargadennec et al., 2022). the existing evidence regarding the clinical utility of PbtO2 monitoring remains contentious. Earlier meta-analyses (Santana et al., 2024; Shanahan et al., 2023; Xie et al., 2017) of both randomized controlled trials (RCTs) and non-randomized controlled trials (NRCTs) demonstrated superior neurological outcomes and decreased mortality rates among STBI patients who underwent combined PbtO2 and ICP monitoring. Conversely, a meta-analysis (Hays et al., 2022) focusing solely on RCTs failed to confirm the beneficial effect of combined monitoring of PbtO2 and ICP on improving neurological outcomes. Moreover, the recently published RCT (Payen et al., 2023) with a substantial sample size indicated that the amalgamation of PbtO2 and ICP monitoring failed to mitigate the proportion of patients experiencing poor neurological outcomes at the six-month mark.

Thus, the aim of the study was to conduct a contemporary meta-analysis of RCTs and NRCTs to evaluating the effects of combined PbtO2 with ICP compared to isolated ICP, examining endpoints such as favorable neurological outcomes at 6 months, mortality rates, and length of stay.

Methods

Study selection

This meta-analysis adhered to the revised PRISMA guidelines (Page et al., 2021) (Supplemental Information 1). The study protocol was pre-registered on the Open Science Framework (https://osf.io/4gfnk). Two investigators autonomously scoured PubMed, Embase, Scopus, and the Cochrane Library until April 10th, 2024, in pursuit of pertinent English articles (search strategies are listed in Supplemental Information 2). The inclusive criteria are as follows: (1) population: adult patients with STBI, defined by admission Glasgow Coma Scale (GCS) ≤ 8; (2) intervention: patients receiving the combination of PbtO2 and ICP monitoring; (3) comparator: patients only receiving isolated ICP monitoring, without PbtO2 monitoring; (4) outcomes: primary outcome was favorable neurological outcome at 6 months, defined as Glasgow Outcome Scale (GOS) (Jennett & Bond, 1975) ≥ 4, or extended Glasgow Outcome Scale (GOSE) (Jennett et al., 1981) ≥ 5, secondary outcomes were in-hospital mortality, long-term mortality (defined as death beyond 6 months), length of stay in intensive care unit (ICU), and length of stay in hospital; (5) design: randomized controlled trials (RCTs) or cohort studies.

Data extraction and quality assessment

To ensure the accuracy and consistency of our data extraction and quality assessment, two independent reviewers (Chengcheng Zhang, and Lingmin Zhou) were involved. Each reviewer independently retrieved and extracted data from the selected studies. Reports considered potential for inclusion were screened in full text. Differences in this process were resolved by consensus including discussing the points of disagreement and reaching an agreement through deliberation. When no consensus was reached, a third co-author (Kai Zhang) would resolve the issue, help ensure that the final data set was agreed upon by all parties, thus maintaining the integrity of the review. Standardized form from the Cochrane Data Collection template was adapted and used to create a study-specific data abstraction form. In order to minimize errors and reduce introduction of potential biases, two authors (Chengcheng Zhang, and Lingmin Zhou) independently extracted data including first author, study period, design, sample size, population characteristics, duration, and outcomes. If an article reported continuous outcomes such as length of stay in ICU or hospital in the form of medians and interquartile ranges (IQR), we applied the methodology advocated by Wan et al. (2014) to transform this data format into mean values and standard deviations (SD). If any pertinent details were missing from the articles, we reached out to the corresponding authors for clarification.

The quality of the included RCTs was evaluated independently by two reviewers (Jian Huang, and Lanxin Cao) utilizing the Cochrane risk of bias tool (Higgins et al., 2011), while cohort studies were evaluated using the Newcastle-Ottawa Scale (Wells et al., 2022). Publication bias was examined through the Egger’s regression test. In instances of discordant assessments, consensus was achieved through a deliberative process involving inclusion of a third adjudicator (Kai Zhang).

Statistical synthesis and analysis

Given the anticipated clinical diversity among the trials included, we employed the random-effects model to compute the risk ratio (RR) with a 95% confidence interval (95% CI) for dichotomous outcomes, and mean difference (MD) with a 95% CI for continuous outcomes. The Higgins inconsistency (I2) statistic was utilized to gauge the variation among studies, where I2 values of <25%, 25 to 75%, and >75% denote low, moderate, and high heterogeneity, respectively (Higgins et al., 2003). We categorized studies based on their design (RCTs versus cohort studies) for subgroup analyses. The sensitivity analysis was conducted to assess the influence of each individual study by excluding them one at a time. All statistical computations were performed using Review Manager 5.3 and Stata 17.0 software. In addition, we performed trial sequential analyses (TSA) by using the TSA software (0.9.5.10 Beta; The Copenhagen Trial Unit, Copenhagen, Denmark) to rule out the possibility of a false positive result (Wetterslev, Jakobsen & Gluud, 2017). The parameters used in TSA were detailed in Supplemental Information 2.

Results

Study selection and study characteristics

Figure 1 presents the flowchart detailing the search and selection procedures. Initially, 1,590 articles were detected. Following the elimination of duplicates and abstract screening, 39 studies were identified, of which 23 were excluded during full-text evaluations, each for specific reasons. Ultimately, our meta-analyses encompassed 16 studies, involving a total of 2,604 patients, with 1,039 patients received the combined monitoring of PbtO2 and ICP, while 1,565 received isolated ICP monitoring. Basic characteristics of included studies is summarized in Table 1. The number of patients ranged from 29 to 629 across studies. Four of included studies (Lee et al., 2010; Lin et al., 2015; Okonkwo et al., 2017; Payen et al., 2023) in the meta-analysis were RCTs, and the rest of 12 studies (Adamides et al., 2009; Barrit et al., 2022; Green et al., 2013; Hoffman et al., 2021; Martini et al., 2009; McCarthy et al., 2009; Meixensberger et al., 2003; Narotam, Morrison & Nathoo, 2009; Roman et al., 2023; Sekhon et al., 2017; Spiotta et al., 2010; Stiefel et al., 2005) were cohort studies. Among the 13 cohort studies, three studies (Adamides et al., 2009; Barrit et al., 2022; Hoffman et al., 2021; Komisarow et al., 2022) used the propensity-matched scores to adjust potential confounders. A total of 11 included studies reported the neurological outcome at 6 months, four studies (Adamides et al., 2009; Lin et al., 2015; Okonkwo et al., 2017; Payen et al., 2023) defined favorable neurological outcome as the GOSE ≥ 5, while the remaining seven studies (Lee et al., 2010; McCarthy et al., 2009; Meixensberger et al., 2003; Narotam, Morrison & Nathoo, 2009; Roman et al., 2023; Sekhon et al., 2017; Spiotta et al., 2010) used the GOS ≥ 4 to define favorable neurological outcome.

Figure 1 PRISMA 2020 flow diagram for this meta-analysis.

Table 1 Characteristics of included studies.

Study	N	Design	Population	Study duration	Management goal	Outcomes	
Payen et al. (2023) (OXY-TC trial)	147/144	Randomized controlled trial	Patients with severe non-penetrating traumatic brain injury, had best pre-hospital GCS ≤ 8 and motor component of 1 to 5	Started within 16 h of brain injury, for 5 days	ICP < 20 mmHg, PbtO2 > 20 mmHg	Favorable outcome at 6 months (GOSE ≥5), in-hospital mortality, long-term mortality	
Roman et al. (2023)	37/40	Retrospective cohort study	Patients had severe TBI, with GCS ≤ 8	Started within 24 h of ICU admission, until status improved	ICP < 20 mmHg, PbtO2 > 20 mmHg	Favorable outcome at 6 months (GOS ≥4), long-term mortality, length of stay in ICU and hospital	
Barrit et al. (2022)	35/35	Retrospective cohort study	Patients with severe TBI, had GCS ≤ 8	Started within 48 h of ICU admission, for 9 days	ICP < 20 mmHg, PbtO2 > 20 mmHg	In-hospital mortality	
Hoffman et al. (2021)	155/310	Retrospective cohort study	Patients with severe TBI, had GCS ≤ 8	Started within 6 h of injury, for 5 days	NR	In-hospital mortality, length of stay in ICU	
Okonkwo et al. (2017) (BOOST-II trial)	57/62	Randomized controlled trial	Patients with non-penetrating severe TBI, had initial GCS ≤ 8 and motor score ≤5	Started within 12 h of injury, for 5 days	ICP < 20 mmHg, PbtO2 > 20 mmHg	Favorable outcome at 6 months (GOSE ≥5), in-hospital mortality, long-term mortality	
Sekhon et al. (2017)	39/76	Retrospective cohort study	Patients admitted to the ICU with a diagnosis of severe TBI (GCS ≤ 8)	NR	NR	Favorable outcome at 6 months (GOS ≥4), length of stay in ICU and hospital	
Lin et al. (2015)	17/19	Randomized controlled trial	Patients with severe TBI, had GCS of 3 to 8	Started within ICU admission, for 5 days	ICP < 20 mmHg, PbtO2 > 20 mmHg	Favorable outcome at 6 months (GOSE ≥5), in-hospital mortality, long-term mortality	
Green et al. (2013)	37/37	Retrospective cohort study	Patients admitted into SICU with severe TBI (GCS ≤ 8)	Until ICP < 20 mmHg and/or PbtO2 > 20 mmHg for 24 h without intervention	ICP < 20 mmHg, PbtO2 > 20 mmHg	In-hospital mortality, length of stay in ICU and hospital	
Lee et al. (2010)	14/15	Randomized controlled trial	Patients with severe TBI, had GCS of 4 to 8	Started within 24 h of ICU admission, for 9 days	ICP < 20 mmHg, PbtO2 > 20 mmHg	Favorable outcome at 6 months (GOS ≥4), in-hospital mortality, length of stay in ICU and hospital	
Spiotta et al. (2010)	70/53	Retrospective cohort study	Patients with severe TBI, had GCS ≤ 8	4.7 days for intervention group and 3.5 days for control group	ICP < 20 mmHg, PbtO2 > 20 mmHg	Favorable outcome at 6 months (GOS ≥4), long-term mortality	
Narotam, Morrison & Nathoo (2009)	127/39	Retrospective cohort study	Patients with major trauma (ISS scores ≥ 16) and TBI (GCS ≤ 8)	NR	NR	Favorable outcome at 6 months (GOS ≥4)	
McCarthy et al. (2009)	81/64	Retrospective cohort study	Patients with severe TBI, defined as GCS ≤ 8	NR	ICP < 20 mmHg, PbtO2 > 20 mmHg	Favorable outcome at 6 months (GOS ≥4), in-hospital mortality, long-term mortality, length of stay in ICU and hospital	
Martini et al. (2009)	123/506	Retrospective cohort study	Patients with severe TBI, had GCS ≤ 8	NR	ICP < 20 mmHg, PbtO2 > 20 mmHg	In-hospital mortality, length of stay in hospital	
Adamides et al. (2009)	20/101	Retrospective cohort study	Patients aged 16 to 70 years with severe TBI, had GCS ≤ 8	Started within 48 h of ICU admission, until status improved	ICP < 20 mmHg, PbtO2 > 20 mmHg	Favorable outcome at 6 months (GOSE ≥5), long-term mortality	
Stiefel et al. (2005)	28/25	Retrospective cohort study	Patients with severe TBI, had GCS ≤ 8	For ≥ 72 h	ICP < 20 mmHg, PbtO2 ≥ 25 mmHg	In-hospital mortality	
Meixensberger et al. (2003)	52/39	Retrospective cohort study	Patients with severe TBI, had GCS ≤ 8	NR	ICP < 20 mmHg, PbtO2 > 1.33 kPa	Favorable outcome at 6 months (GOS ≥4)	
Notes.

GCS Glasgow Coma Scale

ICP Intracranial pressure

PbtO2 brain tissue oxygen pressure

GOSE extended Glasgow Outcome Scale

GOS Glasgow Outcome Scale

TBI Traumatic brain injury

ICU Intensive Care Unit

ISS Injury Severity Score

NR Not reported

The evaluation of bias risk is available in Supplemental Information 3. All RCTs received a high bias risk rating due to their open-label nature. Additionally, all cohort studies attained high quality ratings, scoring above 6 in total.

The publication bias was assessed using Egger’s test and the funnel plot. Evidence of potential publication bias was observed for the favorable neurological outcome and long-term mortality at 6 months (Supplemental Information 4, Egger’s test: P < 0.05). Consequently, an analysis employing the trim and fill method was conducted. Following imputation, the pooled estimate continued to demonstrate improved favorable neurological outcome (RR: 1.30, 95% CI [1.14–1.49]) and reduced long-term mortality (RR: 0.75, 95% CI [0.63–0.91]). Additionally, no significant risk of publication bias was detected for other outcomes (Egger’s test, P > 0.05; Supplemental Information 4).

Primary outcome

A total of 11 studies (four RCTs and seven NRCTs) reported neurological outcome at 6 months. Compared with isolated ICP monitoring, the use of combined PbtO2 and ICP monitoring was associated with a higher favorable neurological outcome rate at 6 months (RR 1.33, 95% CI [1.17–1.51], P < 0.0001, I2 = 0%, Fig. 2). However, the subgroup analysis of RCTs indicated no significant difference in the neurological outcome at 6 months between two groups (RR 1.12, 95% CI [0.91–1.37], P = 0.28, I2 = 0%, Fig. 2). The sensitivity analysis revealed no significant difference in the primary outcome, indicating good robustness (Supplemental Information 4).

Figure 2 Forest plot showing the effect of combined PbtO2 and ICP monitoring versus isolated ICP monitoring on the favorable neurological outcome at 6 months.

Secondary outcomes

A total of 10 studies (four RCTs and six NRCTs) reported the in-hospital mortality, the use of combined PbtO2 and ICP monitoring may reduce the in-hospital mortality, but this did not reach statistical significance (RR 0.81, 95% CI [0.66–1.01], P = 0.06, I2 = 32%, Fig. 3A). The sensitivity analysis did not show any significant difference for the in-hospital mortality (Supplemental Information 4).

Figure 3 Forest plot showing the effect of combined PbtO2 and ICP monitoring versus isolated ICP monitoring on the (A) short-term mortality, (B) long-term mortality.

The long-term mortality rate was reported in nine studies (three RCTs and six NRCTs). Compared with patients received combined PbtO2 and ICP monitoring had lower long-term mortality (RR 0.72, 95% CI [0.59–0.87], P = 0.0008, I2 = 0%, Fig. 3B), whereas the pooled result of RCTs did not show the beneficial effect of combined PbtO2 and ICP monitoring on long-term mortality (RR 0.71, 95% CI [0.37–1.33], P = 0.28, I2 = 56%, Fig. 3B). No significant differences were found in sensitivity analysis for the long-term mortality (Supplemental Information 4).

The analysis of length of stay in ICU and hospital included eight studies (one RCT and seven NRCTs). The pooled results indicated that there was no significant difference between two groups (ICU: MD 2.10, 95% CI [−0.37–4.56], P = 0.10, I2 = 8%, Fig. 4A; hospital: MD 1.07, 95% CI [−2.54–4.67], P = 0.56, I2 = 49%, Fig. 4B). Only one RCT reported the data regarding length of stay, and the length of stay in ICU and hospital was similar between groups. Four studies (Barrit et al., 2022; Green et al., 2013; Hoffman et al., 2021; Sekhon et al., 2017) reported length of stay data in the form of median and IQR. we performed sensitivity analysis by excluding these studies, the results did not show any significant difference for the length of stay in ICU and hospital (Supplemental Information 4).

Figure 4 Forest plot showing the effect of combined PbtO2 and ICP monitoring versus isolated ICP monitoring on the (A) length of ICU stay, (B) length of hospital stay.

An asterisk (*) indicates the study report data in the form of medians and IQR.

Trial sequential analysis

Results of TSA are presented in Supplemental Information 5. For the primary outcome, the TSA showed that the although cumulative Z curve did not reach the required sample size, it had already crossed the conventional boundary and TSA threshold, demonstrating that this finding was conclusive. For the in-hospital mortality, the cumulative Z curve did not reach either the conventional boundary or the TSA threshold, indicating that there was no significant difference between patients received combined monitoring and isolated ICP monitoring. Moreover, the TSA showed that the current meta-analysis did not achieve the required information sizes to detect the pre-specified effect sizes for long-term mortality, length of stay in ICU and hospital, indicating that more trials are required for a definitive conclusion for these outcomes.

Discussion

This meta-analysis compared the combination of PbtO2 and ICP monitoring versus isolated ICP monitoring in patients with STBI. In this study, only results from NRCTs found improvement in favorable neurological outcomes at 6 months and long-term mortality rate in patients with combined monitoring. Based on the limited data from RCTs, no conclusions can be made on beneficial effect of the combined PbtO2 and ICP monitoring for patients with STBI. Therefore, the available evidence supporting the use of combined PbtO2 and ICP monitoring in patients with STBI is limited and of low quality. Moreover, there was no significant difference between two groups in in-hospital mortality rates, length of stay in ICU or hospital. The results of TSA also indicated that more trials are needed to further confirm these findings. There remains the need for powerful evidence from high-quality RCTs with large sample sizes to support the best intensive care monitoring and management of patients with STBI.

While within a neurocritical care unit, ICP monitoring is often viewed as the cornerstone (Hawryluk et al., 2022), evidence suggests that even if the ICP is within the normal range, brain hypoxia might still develop and cause a deterioration in outcome (Eriksson et al., 2012; Maloney-Wilensky et al., 2009). It can be argued that depending exclusively on conventional treatment methods for individuals with TBI might result in prolonged periods of undetected diminished cerebral perfusion. Increasing evidence backs the link between brain hypoxia and unfavorable outcomes in STBI, paving the way for the advancement of PbtO2 therapy (Maloney-Wilensky et al., 2009). The primary objective of maintaining PbtO2 levels is to furnish supplementary insights into the correlation between oxygen availability and demand in the brain. Considering that oxygen deficiency can manifest despite normal ICP levels (Oddo et al., 2011), it is deduced that the compromise in cerebral oxygenation during secondary injuries operates independently from the aforementioned parameters. Within this framework, the comprehensive assessment of cerebral dysfunction, integrating pressure, oxygenation, and brain metabolism monitoring, essentially aims to furnish a comprehensive overview of the patient’s status. Clinically, these assessments facilitate not only the mitigation of cerebral hypoxia but also the customization of treatment, early detection of complications, and consequently, enhancement of clinical outcomes in TBI patients (Meyfroidt et al., 2022).

To our best knowledge, our study is the most up-to-date meta-analysis comparing the outcomes of combined PbtO2 and ICP monitoring versus isolated ICP monitoring in patients with STBI and the first study to assess these data by TSA. By incorporating the recent studies, including the latest RCT (Payen et al., 2023), our results align with and substantiate the outcomes documented in prior meta-analyses (Santana et al., 2024; Shanahan et al., 2023; Xie et al., 2017), further strengthen the evidence that the combined monitoring yielded more favorable neurological outcome and reduced mortality rate at 6 months. Implementing PbtO2 monitoring and addressing low PbtO2 levels alongside conventional ICP-guided management potentially enhances neurological outcomes and long-term survival rates by diminishing the duration of cerebral hypoxia episodes and subsequent secondary brain injury. However, like previous studies, the majority of the data included in our analysis came from NRCTs. A recent meta-analysis of four RCTs suggested that the combined PbtO2 and ICP monitoring cannot improve clinical outcomes including mortality, functional recovery, cardiovascular events or sepsis (Pustilnik et al., 2024). Our subgroup analysis of RCTs indicated the same results, suggesting more high-quality RCTs on neurological outcome after combined PbtO2 and ICP monitoring are needed.

The lack of statistical significance in other parameters, such as in-hospital mortality and length of hospital stay, suggests that the effects of combined monitoring may manifest more prominently over the long term. Recently, TBI has been increasingly recognized not only as an acute ailment but also as a chronic condition with enduring ramifications, including a heightened risk of late-onset neurodegeneration (Wilson et al., 2017). Numerous survivors endure profound disabilities, imposing substantial burdens on both families and society. Consequently, acknowledging TBI as a chronic, progressive condition with lifelong implications further underscores the significance and applicability of all findings.

However, our study demonstrated that patients received combined PbtO2 and ICP monitoring have prolonged stays in ICU, which may increase the risk of ICU-related serious adverse events. Martini et al. (2009) similarly reported a 29% rise in median hospital expenses among individuals undergoing PbtO2 monitoring. This rise may stem from extended ICU stays, potentially leading to prolonged mechanical ventilation duration and a heightened likelihood of developing ventilator-associated pneumonia or acute respiratory distress syndrome. Elevated infection risk and heightened bed occupancy demands consequently escalate healthcare expenses. Furthermore, patients received PbtO2 monitoring may experience more technical failures related to intracerebral catheter, Payen et al. (2023) found more catheter dysfunction and accidental removal in the combined monitoring group.

Limitations

Our study has some limitations and the findings need to be interpreted cautiously. First of all, most of included studies were retrospective cohort studies, which may introduce more biases and confounding factors compared with RCTs. Although results from cohort studies confirmed the combined monitoring was associated with improved neurological outcome and long-term mortality, the pooled results of four RCTs did not find a statistically significant benefit. TSA indicated that more trials are needed to further confirm these findings. Furthermore, variations in patient characteristics and monitoring protocols contribute to the significant diversity observed in hospital and ICU length of stay outcomes. The unequal distribution of participants across groups necessitates a careful interpretation of the findings.

However, these limitations could be addressed by including more high quality RCTs with standardized protocols and outcome measures. Of note, several ongoing RCTs are investigating the impact on clinical outcomes of PbtO2-guided therapy in patients with TBI. The BOOST-III trial (NCT03754114) is the largest RCT enrolling more than 1000 patients with STBI to assess whether the addition of PbtO2 monitor prevents secondary injuries and improves functional outcome (Bernard et al., 2022). The BONANZA-GT (ACTRN12619001328167) is designed to assess the value of PbtO2 monitoring and a stepwise algorithm for detecting impaired cerebral oxygenation in patients with STBI.

Moreover, invasive PbtO2 monitoring solely assesses a narrow cerebral oxygenation region, while STBI exhibit diverse patterns of injury, ranging from diffuse to focal. Uncertainty surrounds the optimal probe placement, though recent RCTs advocate situating the device within unaffected tissue on the opposite side of the most severe injury (Okonkwo et al., 2017; Payen et al., 2023). This positioning aims to provide a more accurate representation of overall oxygenation levels. Similar to other monitoring instruments, these catheters are susceptible to technical and procedural challenges, compromising their reliability. Furthermore, it is imperative to conduct cost-effectiveness analyses to gauge the financial impact of integrating PbtO2 monitoring into clinical settings. Moreover, four studies (Barrit et al., 2022; Green et al., 2013; Hoffman et al., 2021; Sekhon et al., 2017) reported continuous variables in the form of median and IQR, necessitating conversion to mean and SD, a process that potentially introduced bias into our findings.

Conclusion

In conclusion, there was very low-quality evidence to support an improvement in favorable neurological outcomes at 6 months and long-term mortality rate in STBI patients with combined PbtO2 and ICP monitoring. There was no significant difference for other clinical outcomes such as in-hospital mortality rates, length of ICU and hospital stay. Overall, since the available evidence supporting the use of combined PbtO2 and ICP monitoring in patients with STBI is limited and of low quality, further randomized trials with larger sample sizes is warranted to validate these findings and better understand the role of PbtO2 monitoring in TBI management.

Supplemental Information

Supplemental Information 1 PRISMA checklist

Supplemental Information 2 Searching strategies and parameters for trial sequential analysis

Supplemental Information 3 The quality assessment of included studies

Supplemental Information 4 Publication bias assessment by funnel plot and Egger’s test, sensitivity analyses

Supplemental Information 5 Trial sequential analysis

Additional Information and Declarations

Competing Interests

Author Contributions

Data Availability

The authors declare there are no competing interests.

Chengcheng Zhang conceived and designed the experiments, performed the experiments, analyzed the data, prepared figures and/or tables, authored or reviewed drafts of the article, and approved the final draft.

Lingmin Zhou conceived and designed the experiments, performed the experiments, analyzed the data, prepared figures and/or tables, authored or reviewed drafts of the article, and approved the final draft.

Kai Zhang conceived and designed the experiments, performed the experiments, analyzed the data, prepared figures and/or tables, authored or reviewed drafts of the article, and approved the final draft.

Jian Huang performed the experiments, analyzed the data, authored or reviewed drafts of the article, and approved the final draft.

Lanxin Cao performed the experiments, analyzed the data, authored or reviewed drafts of the article, and approved the final draft.

Yuhang Lou performed the experiments, analyzed the data, authored or reviewed drafts of the article, and approved the final draft.

Yushi Fan conceived and designed the experiments, analyzed the data, authored or reviewed drafts of the article, and approved the final draft.

Xinyun Zhang conceived and designed the experiments, performed the experiments, prepared figures and/or tables, authored or reviewed drafts of the article, and approved the final draft.

Yesong Wang performed the experiments, prepared figures and/or tables, authored or reviewed drafts of the article, and approved the final draft.

Wei Cui analyzed the data, authored or reviewed drafts of the article, and approved the final draft.

Lihua Hu conceived and designed the experiments, performed the experiments, analyzed the data, authored or reviewed drafts of the article, and approved the final draft.

Gensheng Zhang conceived and designed the experiments, performed the experiments, analyzed the data, authored or reviewed drafts of the article, and approved the final draft.

The following information was supplied regarding data availability:

The raw measurements are available in the Supplementary Files.

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
