# Peer review of "Brain tissue oxygen pressure combined with intracranial pressure monitoring may improve clinical outcomes for patients with severe traumatic brain injury: a systemic review and meta-analysis"

_PeerJ, doi:10.7717/peerj.18086_

## Round 0.1 · original submission · Major Revisions

Dear Authors,

Please proceed to do major revision to your submitted manuscript. Thank you

Reviewer 1 ·

Basic reporting

I congratulate the authors for conducting a very well-written systematic review and meta-analysis comparing brain tissue oxygen plus intracranial pressure monitoring with intracranial pressure monitoring alone. The authors describe the background of the topic well, and as mentioned in the manuscript, this is not the first meta-analysis on the topic. Although the similar previous study conducted by Santana et al. did not include the OXY-TC trial by Payen et al. and did not include a subgroup analysis of RCTs, a recent meta-analysis of RCTs conducted by Pustilnik et al., which was not mentioned in the manuscript, did include this recent and larger RCT.

Experimental design

The methods used in this meta-analysis are well described both in the manuscript and in the supplementary material, making the study reproducible. However, this study reports that the population of interest is composed only of patients with severe TBI, but the RCT by Lin et al. (2015), which was included in this meta-analysis, included patients with moderate TBI in addition to severe TBI. Furthermore, Table 1 reports that Lin et al. included only patients with severe TBI.

Validity of the findings

Considering the outcomes assessed by the recent meta-analysis of RCTs by Pustilnik et al. and the meta-analysis of RCTs and NRCTs by Santana et al., this meta-analysis does not bring many new findings to the topic, despite having a larger number of patients. The meta-analyses by Pustilnik et al. and Hays et al., which focused solely on RCTs, explored outcomes such as cardiovascular events, pulmonary events, and sepsis, which were not explored in the meta-analysis of both RCTs and NRCTs by Santana et al. Would it be possible to analyze these outcomes in this study? Interestingly, the outcomes of length of ICU stay and length of hospital stay differed from those found by Santana et al. when considering the analyses that include both RCTs and NRCTs. Furthermore, a strength of this study is the trial sequential analysis, which was not explored in the previous meta-analyses mentioned above. Moreover, the authors conducted a thorough analysis of publication bias using funnel plot and Egger's test.

Reviewer 2 ·

Basic reporting

1) BASIC REPORTING

- Thank you for the effort in submitting your manuscript to our journal. This manuscript is a combined systematic review and meta analysis which focus on clinical outcomes of patients who are monitored using a combined ICP and PBTO2.
- The background is well written, however, a brief elaboration should be focused on the pathophysiology of secondary brain insult in STBI.
- The authors also gave good brief summary review on previous RCT in the introduction, focusing on the superiority of combined ICP + PBTO2 on patients’ outcomes ( reduced mortality rates and neurological outcomes). This is a good point showed and should be applauded.
- The usage of good English for this manuscript is commandable and satisfactory. However, I strongly suggest for this manuscript be sent for English editing prior to re-submission, as there were flaws in conjunctions and phrases used in the discussion segment.
- Typing error (I2) was present in line 130 and favourable (line 168).
- The figures attached are clear.

Experimental design

2) EXPERIMENTAL DESIGN

- The PRISMA flow of the study is clear. However, I do not understand the need to have two reviewers to retrieve and extract studies (in the segment of data extraction and quality assessment). Also mentioned there was the need to have a third co-author who will be called in to resolve the issue – May I know what consensus was it revolved and the vital issues that arose? A brief elaboration on the presence of 2 co-authors to retrieve and extract studies should be included.

Validity of the findings

3) VALIDITY OF THE FINDINGS

- The methodology of data analysis are good and strong. However, the is a large data discrepancy in one of the meta analysis by Komisarov et al (2022) which involved 35,501 patients. When I read the detail in the table, however, there were no mention regarding the study’s goal and duration. It was marked as NR. May I know what does NR mean, as it is present in other studies too (for example: Hoffman et al and Sekhon et al). The reason I asked is that whether is it relevant to your study to put the study by Komisarov et al, as the number of patients are too huge as compared to other datas but there were no patient study goals and duration mentioned. This very huge data with no study goals may potentially skew the analysis.
- There are 4 studies which are RCT while the other 13 are retrospective studies. Thus the quality of this manuscript will be jeopardized. I suggest that this manuscript be focused on the four RCT papers, so that the quality is upheld.

Additional comments

4) GENERAL COMMENTS

- Generally, this is a well prepared and written paper. However, there are a few queries pertaining the size of sample patients in one of the studies included.
- May I suggest that this paper be focused on the 4 studies with RCT to maintain the quality.
- One of the studies included 35,501 patients with no study duration and outcome. May I suggest the authors to review and remove it as the data is skewed.

---

## Round 0.2 · accepted · Accept

Congratulations i have checked your submission for both of the prior peer reviewer's comments and concur that revision has been done correctly.

Reviewer 2 ·

Basic reporting

Thank you for the revised manuscript. It certainly looks better now with English editing. In addition the statistical data looks more coherent and doesn't skew.
The discussion is more matured now with this edited manuscript. Congratulations to the team.

Experimental design

The experimental design especially the data interpretation looks good and better now. No extreme skewing is seen now.

Validity of the findings

The findings are valid and doesn't look skewed.

Additional comments

All are perfect. I fully support for this manuscript for publication.

Thank You